# Distortion correction of diffusion weighted MRI without reverse phase-encoding scans or field-maps

**Kurt G. Schilling**[1,2]☯*, **Justin Blaber**[3]☯, **Colin Hansen**[4], **Leon Cai**[5], **Baxter Rogers**[1,2], **Adam W. Anderson**[1,2,4], **Seth Smith**[1,2,4], **Praitayini Kanakaraj**[3], **Tonia Rex**[6], **Susan M. Resnick**[7], **Andrea T. Shafer**[7], **Laurie E. Cutting**[8], **Neil Woodward**[9], **David Zald**[10], **Bennett A. Landman**[1,2,3]

1 Radiology and Radiological Sciences, Vanderbilt University Medical Center, Nashville, TN, United States of America, 2 Vanderbilt University Institute of Imaging Science, Vanderbilt University, Nashville, TN, United States of America, 3 Electrical Engineering, Vanderbilt University, Nashville, TN, United States of America, 4 Computer Science, Vanderbilt University, Nashville, TN, United States of America, 5 Biomedical Engineering, Vanderbilt University, Nashville, TN, United States of America, 6 Vanderbilt Eye Institute, Vanderbilt University Medical Center, Nashville, TN, United States of America, 7 Laboratory of Behavioral Neuroscience, National Institute on Aging, National Institutes of Health, Baltimore, MD, United States of America, 8 Special Education, Vanderbilt University, Nashville, TN, United States of America, 9 Psychiatry and Behavioral Sciences, Vanderbilt University Medical Center, Nashville, TN, United States of America, 10 Neuroscience, Vanderbilt University, Nashville, TN, United States of America

☯ These authors contributed equally to this work.
* kurt.g.schilling@vanderbilt.edu

**Data Availability Statement:** A Singularity virtual machine image has been made available to enable simple evaluation of the proposed techniques at https://github.com/MASILab/Synb0-DISCO. The

## Abstract

Diffusion magnetic resonance images may suffer from geometric distortions due to susceptibility induced off resonance fields, which cause geometric mismatch with anatomical images and ultimately affect subsequent quantification of microstructural or connectivity indices. State-of-the art diffusion distortion correction methods typically require data acquired with reverse phase encoding directions, resulting in varying magnitudes and orientations of distortion, which allow estimation of an undistorted volume. Alternatively, additional field maps acquisitions can be used along with sequence information to determine warping fields. However, not all imaging protocols include these additional scans and cannot take advantage of state-of-the art distortion correction. To avoid additional acquisitions, structural MRI (undistorted scans) can be used as registration targets for intensity driven correction. In this study, we aim to (1) enable susceptibility distortion correction with historical and/or limited diffusion datasets that do not include specific sequences for distortion correction and (2) avoid the computationally intensive registration procedure typically required for distortion correction using structural scans. To achieve these aims, we use deep learning (3D U-nets) to synthesize an undistorted b0 image that matches geometry of structural T1w images and intensity contrasts from diffusion images. Importantly, the training dataset is heterogenous, consisting of varying acquisitions of both structural and diffusion. We apply our approach to a withheld test set and show that distortions are successfully corrected after processing. We quantitatively evaluate the proposed distortion correction and intensity-based registration against state-of-the-art distortion correction (FSL topup). The results illustrate that the proposed pipeline results in b0 images that are geometrically similar to non-

Singularity requires only a b0 and T1 as inputs, and performs all pre-processing (T1 bias field correction and normalization, registration to MNI), image synthesis or model inference, and topup – returning as output topup field coefficients and all intermediate data. Source code and binaries are available at https://github.com/MASILab/Synb0-DISCO.

**Funding:** This work was conducted in part using the resources of the Advanced Computing Center for Research and Education at Vanderbilt University, Nashville, TN. We acknowledge the data provided by several initiatives: ABIDE: Primary support for the work by Adriana Di Martino was provided by the (NIMH K23MH087770) and the Leon Levy Foundation. Primary support for the work by Michael P. Milham and the INDI team was provided by gifts from Joseph P. Healy and the Stavros Niarchos Foundation to the Child Mind Institute, as well as by an NIMH award to MPM (NIMH R03MH096321). DWI Traveling Human Phantom: This work was supported, in part, by awards from CHDI Foundation, Inc.; NIH R01NS050568; NINDS NS40068 Neurobiological Predictors of Huntington's Disease; and NINDS R01 NS054893 Cognitive and Functional Brain Changes in Preclinical HD HCP: Data were provided [in part] by the Human Connectome Project, WU-Minn Consortium (Principal Investigators: David Van Essen and Kamil Ugurbil; 1U54MH091657) funded by the 16 NIH Institutes and Centers that support the NIH Blueprint for Neuroscience Research; and by the McDonnell Center for Systems Neuroscience at Washington University. This work was supported by the National Institutes of Health under award numbers R01EB017230, and T32EB001628, and in part by ViSE/VICTR VR3029 and the National Center for Research Resources, Grant UL1 RR024975-01, and Department of Defense award number W81XWH-17-2-055. This research was conducted with the support from Intramural Research Program, National Institute on Aging, NIH. The content is solely the responsibility of the authors and does not necessarily represent the official views of the NIH. We gratefully acknowledge the support of NVIDIA Corporation with the donation of the Titan Xp GPU used for this research.

**Competing interests:** This work was conducted in part using the resources of the Advanced Computing Center for Research and Education at Vanderbilt University, Nashville, TN. This work was supported by the National Institutes of Health under award numbers R01EB017230, and T32EB001628, and in part by ViSE/VICTR VR3029 and the National Center for Research Resources,

distorted structural images, and more closely match state-of-the-art correction with additional acquisitions. In addition, we show generalizability of the proposed approach to datasets that were not in the original training / validation / testing datasets. These datasets included varying populations, contrasts, resolutions, and magnitudes and orientations of distortion and show efficacious distortion correction. The method is available as a Singularity container, source code, and an executable trained model to facilitate evaluation.

## 1. Introduction

The rapid echo planar imaging techniques and high gradient fields typically used for diffusion weighted magnetic resonance imaging (DW-MRI) introduce geometric distortions in the reconstructed images. Initially, both the static field distortions (e.g., interactions of fast imaging techniques with inhomogeneities) and the gradient dependent effects (e.g., gradient field disturbances given eddy current effects) were corrected along with motion through registration [1, 2]. However, image-based approaches have two central problems. First, accurate inter-modality alignment between (distorted) DW-MRI imaging and (undistorted) T1w anatomical imaging is problematic, especially in areas with limited tissue contrast. Second, image registration does not offer a mechanism for correcting signal pileup–areas of erroneous signal void and/or very bright signal. Modern approaches resolve these difficulties by acquiring additional information, either with a field map or supplementary diffusion acquisitions designed to be differently sensitive to susceptibility and eddy effects (so called "blip-up blip-down" designs). Field maps are effective, but offer limited robustness to acquisition artifacts [3], and the blip-up/blip-down studies are widely used, including in the Human Connectome Project [4].

Current tools such as FSL's topup [5] and TORTOISE [6], use minimally weighted DW-MRI images acquired with different phase-encoding parameters to estimate the static susceptibility field maps. Then, a subsequent pass uses the diffusion weighted images to model and correct for the eddy current effects (e.g., FSL's eddy [7] and TORTOISE's DR-BUDDI [8]). Techniques and datasets for benchmarking [9, 10] and quality control [11] are actively being explored, as obtaining a sufficiently high quality ground truth that is generalizable to clinical studies is difficult. Moreover, there is active research on correction techniques for DW-MRI outside of the brain, e.g., prostate [12] and spinal cord [13].

Despite the availability of effective tools, the supplementary information necessary for these techniques is not always available, which could be potentially due to scanner limitations, scan time constraints, acquisition difficulties / artifacts, or legacy considerations. Recently, we presented a deep learning synthesis approach, Synb0-DisCo, to estimate non-distorted (infinite bandwidth) minimally weighted images from T1 weighed (T1w) images [14]. Synb0-DisCo uses a 2.5D (multi-slice, multi-view) generative adversarial network (GAN) to perform the image synthesis process.

While Synb0-DisCo is a promising first approach for a deep learning solution to the DW-MRI distortion correction problem, it has several limitations. First, Synb0-DisCo does not intrinsically compensate for absolute intensities of the target minimally weighed scans, and therefore, secondary adjustment of the intensity spaces is needed. Second, patient specific contrasts seen in the acquired distorted DW-MRI cannot be learned as the network only had relatively homogeneous T1w MRI information available. Third, Synb0-DisCo is susceptible to 3D inconsistencies as the model did not have access to full imaging context.

Herein, we propose a second generation of our deep learning approach, termed Synb0, for DW-MRI distortion correction to address these limitations. Briefly, we (1) generalize the

Grant UL1 RR024975-01, and Department of Defense award number W81XWH-17-2-055. This research was conducted with the support from Intramural Research Program, National Institute on Aging, NIH. The content is solely the responsibility of the authors and does not necessarily represent the official views of the NIH. We gratefully acknowledge the support of NVIDIA Corporation with the donation of the Titan Xp GPU used for this research, and this does not alter our adherence to PLOS ONE policies on sharing data and materials.

learning approach to use both T1w and distorted DW-MRI images in order to synthesize a b0 with both appropriate geometry and contrast, (2) redesign the network to use full 3-D information, and (3) train across a much larger collection of patients / studies / scanners in order to facilitate generalization across different datasets with varying acquisitions and cohorts. We evaluate Synb0 using three unique datasets with varying image quality, contrast, and acquisitions with baseline consideration of image registration, Synb0-DisCo, and no correction relative to the best available techniques using supplementary acquisitions. Finally, we show generalizability of this methodology by applying the proposed method to eleven different open-sourced datasets that were not included in the training nor testing data.

## 2. Materials and methods

The high-level overall pipeline is shown in Fig 1. The aim is to synthesize an *undistorted* b0 from an input *distorted* b0 and a T1 anatomical image. Using the topup setting of an infinite bandwidth will correct for known deformations and movement to match the undistorted image as well as possible and provide the necessary estimations to proceed with eddy current correction (e.g., with FSL's eddy). Put another way, when application of topup and its advanced distortion correction features and assumptions would traditionally not be possible (due to the absence of reverse PE scans or field maps), we are synthesizing a geometrically *undistorted* image in order to provide topup the information necessary to correct the *distorted* diffusion data.

### 2.1 Data

The data used for this study were retrieved in de-identified form from the Baltimore Longitudinal Study of Aging (BLSA), Human Connectome Project (HCP), and Vanderbilt University. All human datasets from Vanderbilt University were acquired after informed consent under supervision of the appropriate Institutional Review Board. All additional datasets are freely available and unrestricted for non-commercial research purposes. This study accessed only de-identified patient information. Importantly, these datasets have varying resolutions, signal-to-noise ratios, T1 and diffusion contrasts, magnitudes of distortions, and directions of distortions.

Briefly, BLSA acquisition included T1-weighted images acquired using an MPRAGE sequence (TE = 3.1 ms, TR = 6.8 ms, slice thickness = 1.2 mm, number of Slices = 170, flip angle = 8 deg, FOV = 256x240mm, acquisition matrix = 256×240, reconstruction

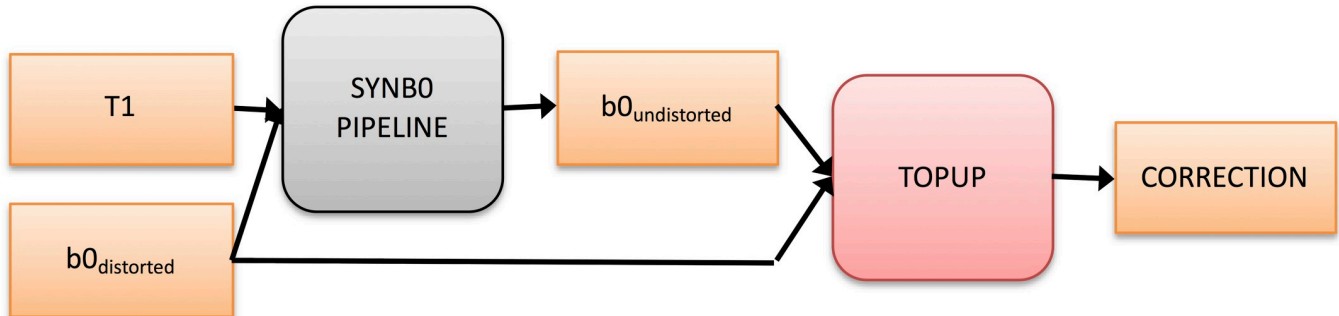

**Fig 1. Overall pipeline.** The goal is to generate an undistorted b0 from a single blip (distorted) b0 and an anatomical T1 image through a deep learning approach. The undistorted image can then be concatenated with the distorted b0 and run through FSL's topup using a simulated infinite PE-bandwidth. This final correction can be used with FSL's eddy (or another eddy current modeling tool) to provide a full correction for diffusion data given only a single phase encoding. Note that the proposed algorithm does not seek to model/correct eddy current effects.

matrix = 256×256, reconstructed voxel size = 1x1mm). Diffusion acquisition was acquired using a single-shot EPI sequence, and consisted of a single b-value (b = 700 s/mm$^2$), with 33 volumes (1 b0 + 32 DWIs) acquired axially (TE = 75 ms, TR = 6801 ms, slice thickness = 2.2 mm, number of slices = 65, flip angle = 90 degrees, FOV = 212*212, acquisition matrix = 96*95, reconstruction matrix = 256*256, reconstructed voxel size = 0.83x0.83 mm). HCP acquisition included T1-weighted images acquired using an 3D MPRAGE sequence (TE = 2.1 ms, TR = 2400 ms, slice thickness = 0.7 mm, flip angle = 8 deg, FOV = 224x224mm, acquisition, voxel size = 0.7x0.7mm). Diffusion acquisition was acquired using a single-shot EPI sequence, and consisted of three b-values (b = 1000, 2000, and 3000 s/mm$^2$), with 90 directions (and 6 b = 0 s/mm$^2$) per shell (TE = 89.5 ms, TR = 5520 ms, slice thickness = 1.25 mm, flip angle = 78 degrees, FOV = 210*180, voxel size = 1.25mm isotropic). The scans collected at Vanderbilt were part of healthy controls from several projects a typical acquisition is below, although some variations exist across projects. T1-weighted images acquired using an MPRAGE sequence (TE = 2.9 ms, TR = 6.3 ms, slice thickness = 1 mm, flip angle = 8 deg, FOV = 256x240mm, acquisition matrix = 256×240, voxel size = 1x1x1mm). Diffusion acquisition was acquired using a single-shot EPI sequence, and consisted of a three b-values (b = 1000, 2000, 3000 s/mm$^2$), with 107 volumes (11 b0 +96 DWIs per shell) acquired axially (TE = 101 ms, TR = 5891 ms, slice thickness = 1.7 mm, flip angle = 90 degrees, FOV = 220*220, acquisition matrix = 144*144, voxel size = 1.7mm isotropic). We again note that variations in acquisition parameters exist in this dataset (resolution up to 2.5mm isotropic).

The data for training the network consists of T1 and distorted b0 image inputs and a truth of undistorted b0 images. For HCP and Vanderbilt, the undistorted b0 images were obtained by running topup on opposite phase encoded b0 images. For HCP, these phase encodings were L-R while for Vanderbilt, the phase encoding were A-P. For BLSA, the undistorted b0 images were obtained using a multi-shot EPI acquisition. The distorted b0 images from BLSA have a phase encoding along the A-P direction. Qualitative depictions of the data (T1, distorted, and undistorted processed b0's) are shown in Fig 2, while the number of datasets and scan information are shown in Table 1.

## 2.2 Preprocessing

The first step for preprocessing was a special step needed for the BLSA data because the intensities for the distorted b0 and undistorted b0 were not guaranteed to match due to the fact that the undistorted b0 was a separate acquisition with a potentially different gain factor. To account for this, the median value of the masked undistorted b0 was scaled such that it matched the masked median value of the undistorted b0 (in a process similar to that done by topup with the *-scale* option). The rest of the data had undistorted b0s computed from topup, which have the same intensities as the distorted image. The rest of the preprocessing steps were applied to the rest of the data in the same manner.

A summary of the preprocessing is shown in Fig 3. The inputs are the T1 image, the distorted b0, and the undistorted b0, while the outputs are a normalized T1, and distorted and undistorted b0, all registered and transformed to MNI-space. To do this, the T1 image was intensity normalized using FreeSurfer's *mri_nu_correct*, *mni*, and *mri_normalize* which perform N3 bias field correction and intensity normalization, respectively on the input T1 image [15]. Next, the distorted b0 and undistorted b0 were coregistered to the skullstripped (via *bet*) T1 using FSL's *epi_reg* [2] (a rigid-body 6 degrees of freedom transformation). The T1 was then affine registered using ANTS to a 1.0 mm isotropic MNI ICBM 152 asymmetric template [16]. The FSL transform from *epi_reg* was converted to ANTS format using the *c3d_affine_tool*

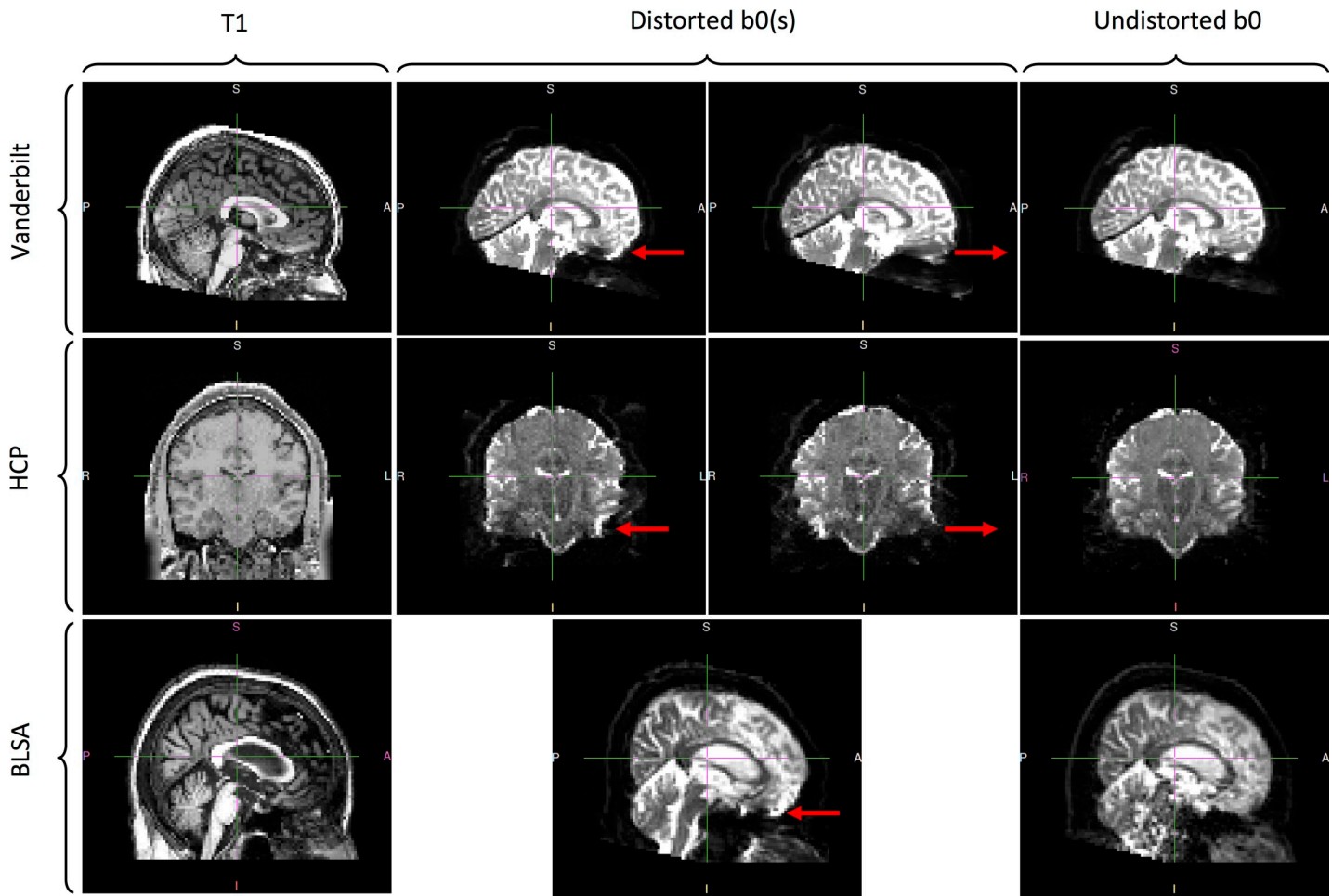

**Fig 2. Datasets used in this study.** The b0's from Vanderbilt were acquired with opposite phase encodings along the A-P direction and corrected with topup. The b0s from HCP were acquired with opposite phase encodings along the L-R direction and corrected with topup. Lastly, b0s from BLSA were acquired with a single phase encoding along the A-P direction and corrected via a multi-shot EPI acquisition. The arrows in the distorted b0 columns highlight areas of visible susceptibility distortion.

and the b0s were transformed into 2.5 mm isotropic MNI space via *antsApplyTransforms*. All transforms were saved so the inverse transform could be applied to bring the results back into

**Table 1.**

| Learning and Testing | | Vanderbilt | HCP | BLSA |
|---|---|---|---|---|
| | Subjects | 38 | 488 | 424 |
| | Sessions | 80 | 488 | 529 |
| | Phase Encoding | A-P | L-R | A-P |
| | Correction | Topup | Topup | Multishot EPI |
| | Resolution (mm) | 1.7–2.5 iso | 1.25mm iso | 0.83x0.83x2.2 |
| | TE/TR | 101/5891 | 89.5/5520 | 75/6801 |
| | Training splits (subjects) | | | |
| | Learning (Training + Validation) | 35 | 433 | 381 |
| | Testing (with-held) | 3 | 55 | 43 |

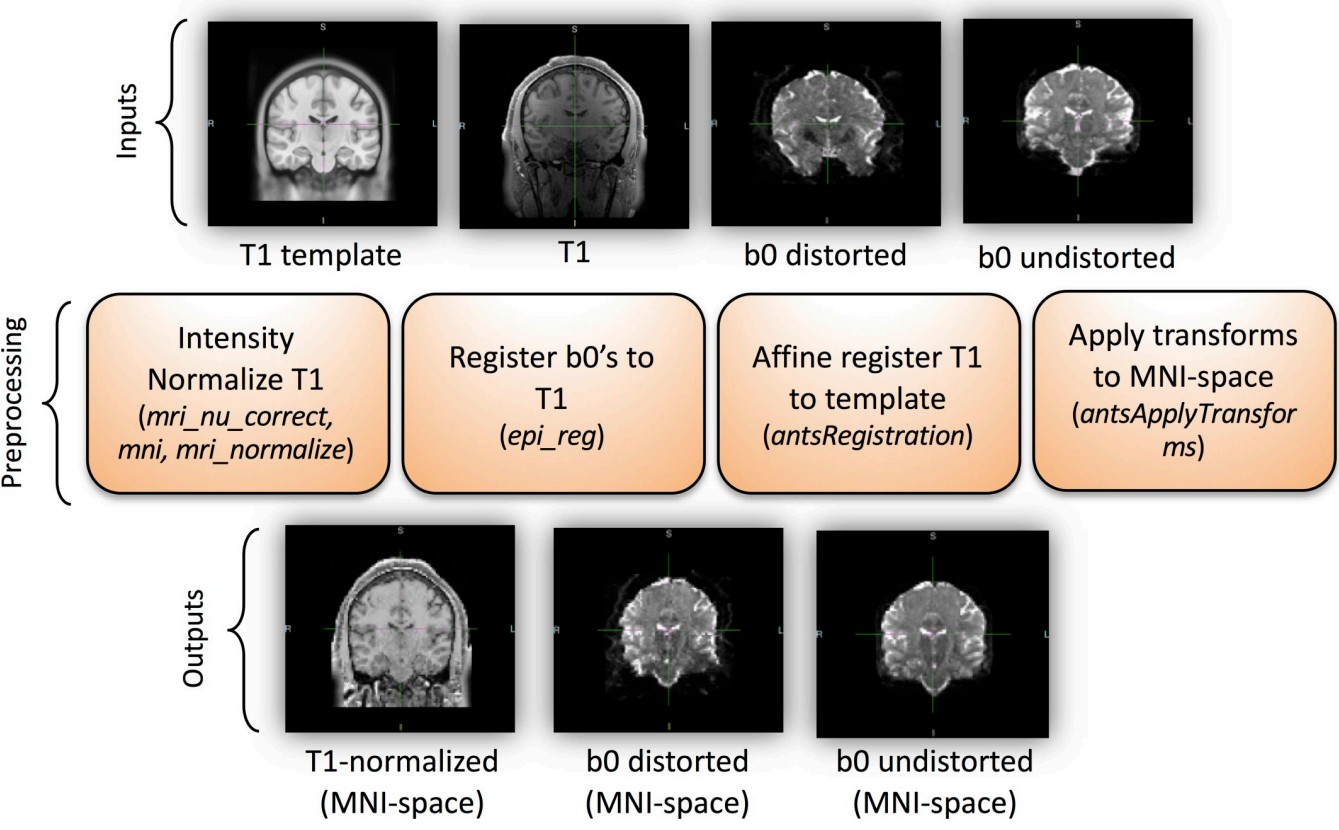

**Fig 3. The preprocessing pipeline.** This figure show data preparation prior to network learning (Fig 4). The pipeline inputs includes a T1 image as well as distorted and undistorted b0 images, and the outputs are all images aligned in MNI space.

subject space. Additionally, whole volume masks were created for the undistorted b0, distorted b0, and T1 and transforms were applied as needed to these masks to prevent training on regions where resampling could not be done.

Before training, the normalized 2.5 mm atlas aligned T1's intensities were linearly scaled such that intensities ranging from 0 to 150 were mapped between -1 and 1. Fixed values of 0 and 150 could be used because of the FreeSurfer T1 intensity normalization as described. The distorted b0's intensities were scaled such that 0 to the 99$^{th}$ percentile were mapped between -1 and 1. Using the min and max of the distorted b0 was unstable due to signal pileup (which can cause localized large values). The 99$^{th}$ percentile was close enough to get the intensity of the cerebrospinal fluid mapped to 1. For the undistorted b0, the same 99$^{th}$ percentile value found for the undistorted image was used to scale it between– 1 and 1. This was to ensure the same scaling was applied for the distorted and undistorted b0 since their overall intensities should be the same.

## 2.3 Network/training/loss

The network, inputs and outputs, and loss calculation are diagrammed in Fig 4. The network used to generate the undistorted b0 in 2.5 mm space was a 3D U-Net [17, 18] (2 channel input and 1 channel output), based on the original implementation in PyTorch [19]. Some differences were that leaky ReLU were used in place of ReLU. In addition, instance norm was used

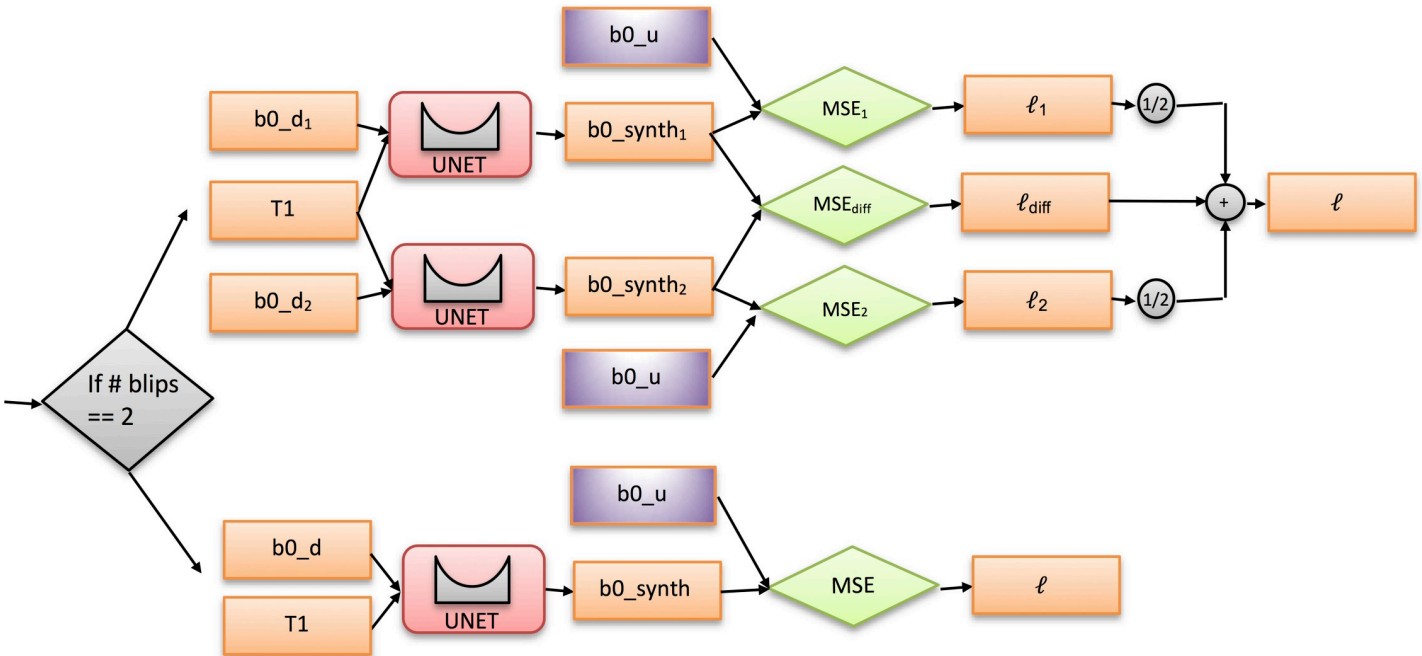

**Fig 4. Training logic and loss calculation.** Each U-net has 2 input channels (a T1 image and distorted b0) and a single output channel (synthesized undistorted b0). For single blip b0's (BLSA; lower half of the decision tree), only the "truth" loss was computed. For two blip b0's (HCP and Vanderbilt; upper half of decision tree), two "truth" losses were computed, averaged, and then a "difference" loss term was added to obtain the final loss.

in place of batch norm since a small batch size was used. The implementation is available within a singularity container release (https://www.singularity-hub.org/collections/3102).

For training purposes, the data (organized in BIDS format [20]) was partitioned across subjects for the test/validation/training sets. The data set was first partitioned into a test set of 100 random subjects and a "learning" set of 850 subjects. The test set was completely withheld. The "learning" set was again partitioned using 5-fold cross validation into training and validation sets (i.e., randomly shuffled into 680 testing and 170 validation for each fold).

The network trained for 100 epochs with a learning rate of 0.0001. Adam optimizer was used with betas set to 0.9 and 0.999. A weight decay of 1e-5 was applied. For each fold, the network was trained and after each epoch, the validation mean squared error (MSE) was computed and stored. The network with the lowest validation was selected for each fold as the most optimal network, resulting in 5 trained networks. Training was performed on Nvidia TITAN Xp GPUs with 12 GB of memory.

The loss function depended on the input data. For the BLSA subjects, since there was only single blip b0s (a distorted b0, $b0\_d$), the output of the U-Net (a synthesized b0, $b0\_synth$) was compared directly to the undistorted image ($b0\_u$) with MSE to generate the loss ($\ell$). For HCP and Vanderbilt images, there were two blip b0s ($b0\_d_1$ and $b0\_d_2$). Both distorted b0s were passed through the network. Both outputs ($b0\_synth_1$ and $b0\_synth_2$) were compared with the undistorted b0 ($b0\_u$) with MSE ($MSE_1$ and $MSE_2$) and the average of the two was stored as the "truth" loss ($\ell_1 + \ell_2 / 2$). In addition, the two outputs ($b0\_synth_1$ and $b0\_synth_2$) were subtracted and compared via MSE loss ($MSE_{diff}$), which we consider as the "difference" loss ($\ell_{diff}$). These two losses were summed to get the final loss ($\ell$). The "truth" loss can be interpreted as minimizing the bias of the result (output should not deviate far from the truth). The "difference" loss can be interpreted as minimizing the variance of the result (outputs should be the same). For all losses computed, masks were used as described in the preprocessing section to

only compute the loss in regions where resampling could be done. This strategy, of including both single blip data, as well as two-blip data, let's the networks learn from distortions in a number of directions. This network architecture mirrors the Siamese [21] and null space [22] network designs.

## 2.4 Pipeline

For the final network, the five networks trained during cross validation were used and the ensemble average of the result was taken to get the synthesized undistorted b0 in affine MNI space. The inverse transforms were used to warp the generated undistorted b0 back into subject space. The distorted b0 was smoothed slightly to match the smoothness of the undistorted b0 because the output resolution of 2.5 mm isotropic from the network, followed by resampling back to subject space, resulting in some smoothing due to interpolation in the undistorted b0. We believe this is only required due to the fact that we are constrained to 2.5 mm isotropic for the network input due to GPU memory, so this step would be unnecessary if a higher resolution network on a GPU with more memory was used. The slightly smoothed distorted b0 and undistorted b0 were merged together and passed into topup with an acquisition parameters file containing two rows (see overall pipeline in Fig 1). The first row is a "dummy" row with arbitrary readout set, although care must be taken to ensure the arbitrary value is set in the correct column depending on the phase encoding direction. The second row is set with a readout of time of 0, which lets topup know that the second volume (the undistorted b0) contains no susceptibility distortion. The end result is the correction from topup which can be used as input into FSL's eddy to perform a full diffusion imaging pre-processing which includes distortion, eddy current, and motion correction.

## 2.5 Quantitative evaluation with cross-validation

To quantitatively investigate the geometric fidelity and contrasts of the images from the proposed pipeline, the resulting b0 images were compared to both the (undistorted) T1 image and the state-of-the art distortion correction (topup). For each of the classes of data (i.e., Vanderbilt, HCP, BLSA), 5 subjects were randomly selected from the withheld test set (i.e., had never been used in any part of the model selection process), resulting in 15 evaluations. For each, two measures were calculated. First, mutual information (MI) of the b0 with T1 was calculated as a measure of geometric similarity. Second, the mean-squared error of signal intensities between the synthesized correction and the topup correction is calculated, which assesses both distortion correction accuracy and contrast accuracy. For comparison, these measures were calculated with (A) the uncorrected b0, (B) a standard registration-based distortion correction method (that from Bhushan et al., 2012 [23] implemented using the default parameters in BrainSuite software toolkit [24]), and (C) the output from the proposed synthetic distortion correction (note that the synthesized b0 is not used for comparison, rather the acquired b0 after distortion correction is used for quantitative analysis).

## 2.6 Quantitative evaluation with external validation

We additionally chose a number of external validation datasets (not used in testing, training, nor validation steps) in order to validate our algorithm on data from sets entirely different from testing/training/validation. All datasets are freely available and unrestricted for non-commercial research purposes (and found through literature searches, searches through https://openneuro.org, or through https://www.nitrc.org). These include the "MASSIVE" brain dataset [25], Kirby21 dataset [26], the age-ility project dataset, an ABIDE dataset (ABIDE I) [27], IXI datasets (acquired at both Hammersmith hospital and Guys hospital)(https://brain-

development.org/ixi-dataset/), SCA2 DTI dataset [28], a "DWI Traveling Human Phantom" dataset [29], HCP Lifespan data (HCP Development [30], subject age = 8), MGH HCP dataset [31], and a Unilateral Glaucoma 3T dMRI dataset (dataset doi: 10.18112/openneuro.ds001743. v1.0.1). Importantly, these are all acquired with widely varying acquisition conditions for both T1 and diffusion images, on different scanners, different resolutions, with different contrasts and levels of distortions. For example, the MASSIVE dataset [25] was acquired on one subject over 18 sessions (T1 acquired at 1mm isotropic resolution using a 3D-TFE sequence, diffusion acquired at 2.5mm isotropic resolution, TE = 100ms, TR = 7000, flip angle = 90, PE = AP direction), Kirby 21 [26] is a scan-rescan reproducibility dataset (T1 acquired at 1x1x1.2mm resolution using a MPRAGE sequence, diffusion acquired at 2.2mm isotropic resolution, TE = 67ms, TR = 6281, flip angle = 90, PE = AP direction). As a final example, Age-ility [32] is a project that aims to investigate cognition and behavior across the lifespan (T1 acquired at 1mm isotropic resolution using a MPRAGE sequence, diffusion acquired at 2mm isotropic resolution, TE = 108ms, TR = 15,300, flip angle = 90, PE = AP direction). We refer to the appropriate references for detailed acquisitions descriptions of each dataset.

## 3. Results

### 3.1 Results with cross-validation

The resulting training/validation and test results are shown in Fig 5. There are 5 validation curves (dashed lines) and 5 training curves (solid lines) since 5-fold cross validation was used. Note that the test MSE falls within the same range of the tail end of the training/validation curves. Training took 2.5 days to complete on a single Nvidia TITAN Xp GPU.

Fig 6 shows results from the withheld test set, including distorted b0 (indicated by a "D") and corrected (or undistorted) b0 after application of the proposed distorted correction (indicated by a "U"), for the Vanderbilt, HCP, and BLSA datasets. Note that the corrected b0 in Fig 6 represents the results of the entire proposed pipeline–synthesizing an undistorted b0, then

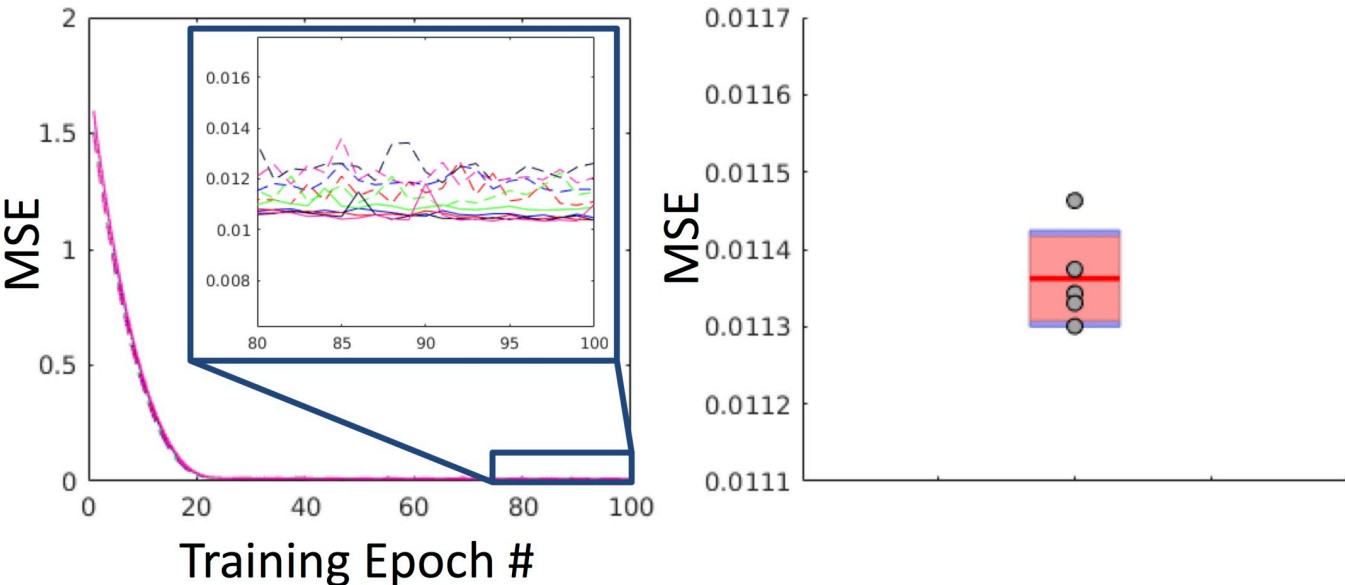

**Fig 5. Training, validation, and withheld loss.** Left: Training and validation curves for each fold (5 training loss curves and 5 validation loss curves). The solid lines are the training curves and the dashed lines are the validation curves. Right: Plot of the MSE of the withheld test set (N = 100) for each fold shown as gray dots (5 folds) against a boxplot of the tail-end of the validation curves for each fold. Note that the test loss falls within the same range of the tail-end of the validation curves.

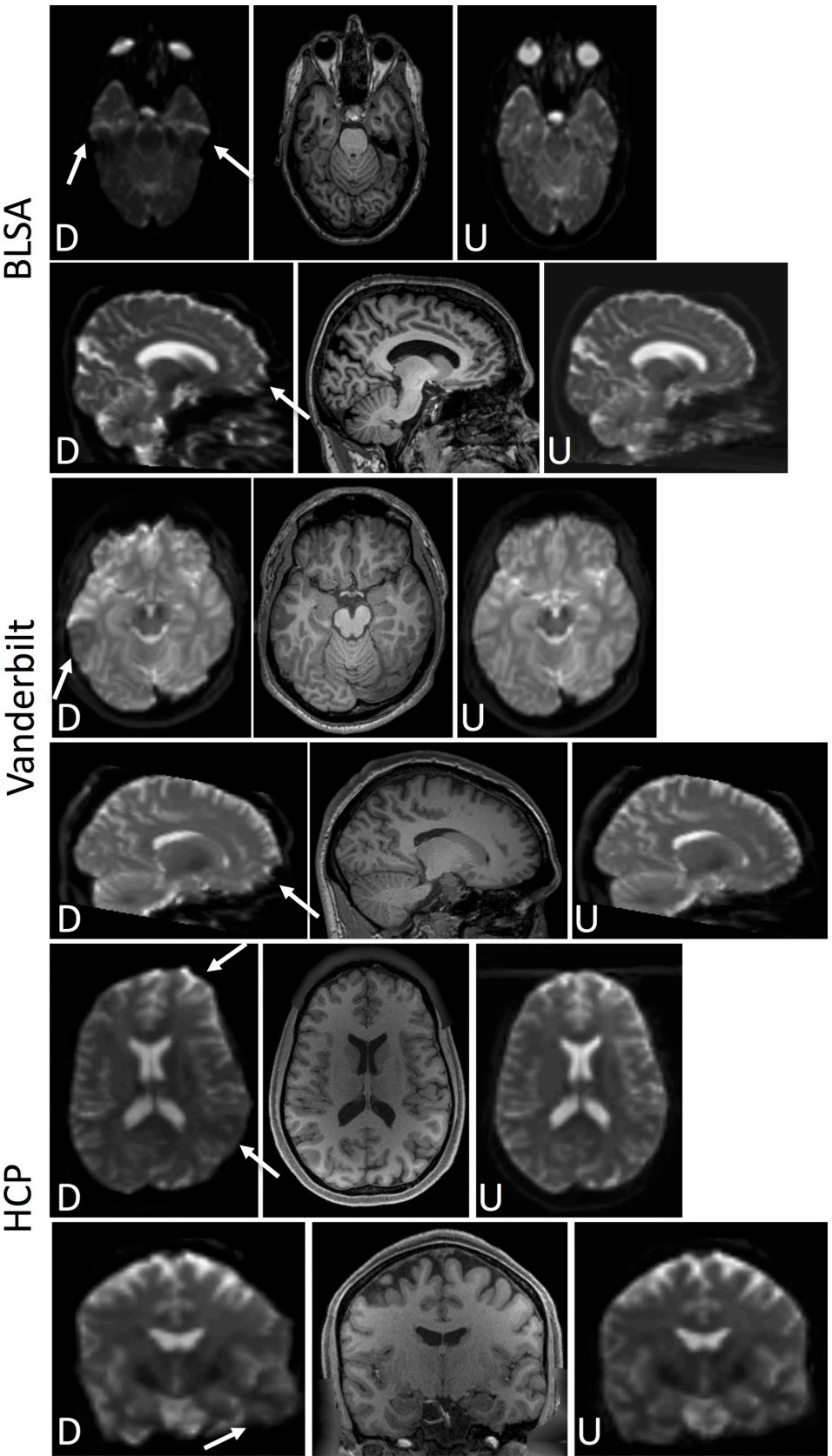

**Fig 6. Withheld test set results.** BLSA (top), Vanderbilt (middle), and HCP (bottom) datasets, the distorted ("D") and undistorted (after applying the proposed pipeline) b0 ("U") are displayed along with a structurally-undistorted T1 image. This demonstrates qualitatively improved alignment to the subjects' T1 using the proposed pipeline (i.e., synthesized b0 and topup correction). Arrows highlight areas of observable improvement as described in the text.

applying topup to the synthesized images. Thus, we are visualizing *corrected* b0 images and not the *synthesized* images.

In all cases, it is clear that the corrected b0 is geometrically more similar to the T1 image than uncorrected, indicating significant reductions in distortions. For Vanderbilt and BLSA data, the most pertinent region of correction is the anterior region of the brain, and mid-brain areas. For HCP, left/right distortion is clearly corrected, and is most obvious in the temporal lobe and inferior aspects of the white/gray matter boundary.

To verify anatomically faithful distortion correction, it is critical to quantify geometric similarity of the resulting corrected b0 images to the co-registered (and undistorted) T1. Fig 7 (left) shows the mutual information between a non-corrected b0 (N.C.), registration-corrected b0 (R.C.), the legacy synthetic distortion (S.D.) [33], and the proposed synthetic-correction (Syn. C.), where a higher value serves as an indicator of a closer match to the structural scan. It is clear that all correction methods significantly improve brain geometry. We point out that the legacy distortion correction [33] (S.D.) appears to show improvements mainly for the BLSA datasets (which it was trained on) and did not well generalize to additional contrasts and geometries–limitations which the currently proposed approach specifically intend to address. Fig 7 (right) quantifies the MSE of each b0 with the state-of-the art topup-corrected b0. In this case, the synthesized method shows significant improvements in both geometry and contrast (with one outlier). Thus, results are structurally similar to T1, and on par with registration techniques (as assessed by MI to T1) and more closely match the ground truth state-of-the art topup correction (as assessed by MSE with TOPUP b0).

## 3.2 Results with external validation

We apply the proposed synthesis+topup pipeline using data from existing open-sourced diffusion datasets that were not included in training (Table 1). Fig 8 shows that this pipeline can correct distortions on datasets that may differ from those the networks were trained on. Specifically, we use the MASSIVE, Age-ility, and Kirby21 datasets, all of which are acquired at

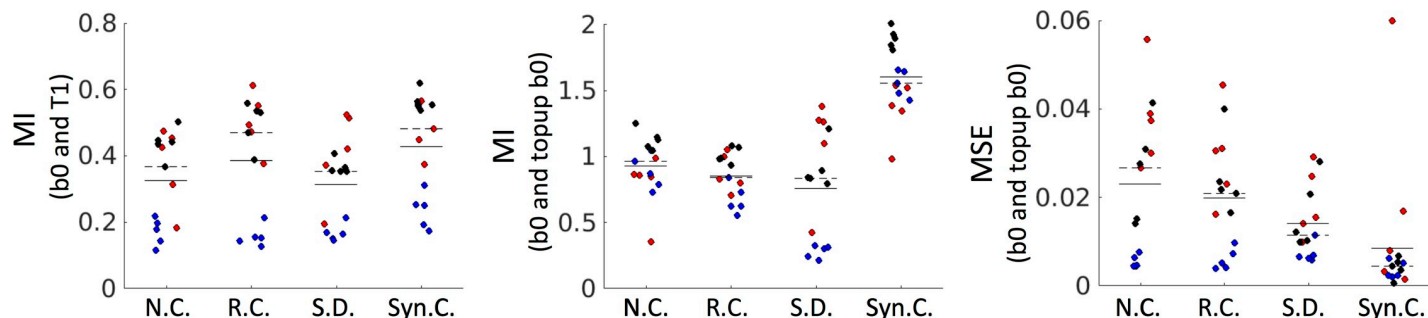

**Fig 7. Validation of geometry and contrast after distortion correction.** Top: MI of the non-corrected (N.C), registration corrected (R.C.), legacy synthetic-distortion (S.D) and proposed synthetic correction (Syn.C.) b0 images with the structural T1 image. A higher value suggests a geometry more similar to the undistorted T1. MIddle: MI of the N.C., R.C., S.D, and Syn.C. b0 with state-of-the art topup distortion correction results. A higher value indicates geometry/contrast more similar to the goldstandard. Bottom: MSE of the N.C., R.C., S.D, and Syn.C., b0 with state-of-the art topup distortion correction results. A lower value indicates structure and image intensities more similar to the topup results. For both, solid and dashed lines indicate mean and median values, respectively. Each contains 15 datapoints, from 5 HCP subjects (blue), 5 Vanderbilt subjects (black), and 5 BLSA subjects (red).

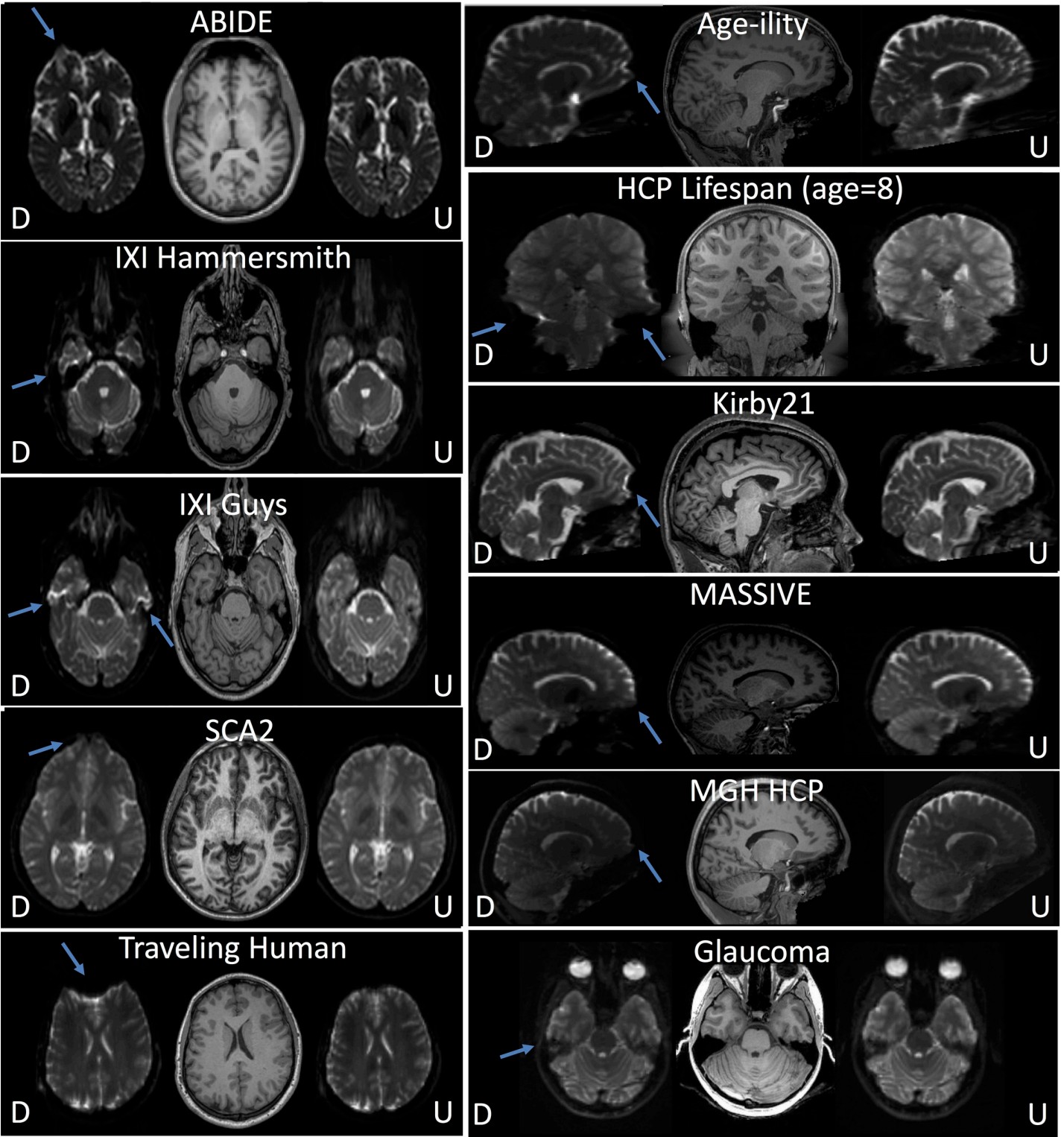

**Fig 8. External dataset validation.** External validation of corrected b0's after applying the synthesized b0 distortion correction pipeline with data from open-sourced studies. The distorted ("D") and undistorted ("U") b0 images are shown alongside T1 images. In all cases, effective distortion correction is visually apparent (distortions indicated by arrows).

varying resolutions, different distortion directions, different brain sizes, and different subject ages. Most areas show significantly improved geometric match to T1's, for example frontal areas, ventricles, and brainstem indicating effective distortion correction. Quantifying MI with T1 as a proxy for geometric similarity shows statistically significant improvement in correction (paired t-test, p<0.001), and an increased MI for all 11 samples tested (MI distorted: 0.42 ±0.98; MI undistorted: 0.53±0.13).

## 4. Discussion

The Synb0 substantively improves upon the state-of-the art for distortion correction of DW-MRI data without supplementary acquisitions. Synb0 more accurately identifies anatomical geometry than image-based distortion correction as assessed by mutual information and mean squared error. The improvement is consistent across multiple datasets. Moreover, Synb0 runs in ~2 minutes per scan (specifically, inference, or generation of synthetic images, is ~2 minutes), versus ~10–15 minutes for image-based registration. It is important to point out that the full proposed pipeline still involves running topup, which can vary from ~20–40 minutes depending on image resolution and topup configuration.

We emphasize that correction without modern/supplementary sequences is not a first choice for study design. However, vast quantities of DW-MRI have been acquired (and are still being acquired) with classic/limited DW-MRI sequences (e.g., legacy studies, older scanners, scanners without advanced DW-MRI license keys, clinically acquired imaging). Hence, it is important to have the best possible alternative processing strategies for these data.

This effort is the second publication to examine deep learning for DW-MRI distortion correction. Mutual information is improved by a mean of 36% over the prior publication and 11% over registration correction (Fig 7A and 7B). On a study by study basis, these are statistically significant (p<0.001, paired t-test) across all individual cohorts. Similarly, mean squared error is improved (decreased) by a mean of 40% over the prior publication and 63% over registration correction (Fig 7C), with differences in cohorts showing statistical significance (p<0.001, paired t-test).

A Singularity virtual machine image has been made available to enable simple evaluation of the proposed techniques at https://github.com/MASILab/Synb0-DISCO. The Singularity requires only a b0 and T1 as inputs, and performs all pre-processing (T1 bias field correction and normalization, registration to MNI), image synthesis or model inference, and topup–returning as output topup field coefficients and all intermediate data. Source code and binaries are available at https://github.com/MASILab/Synb0-DISCO. These open source efforts simplify training or transfer learning with larger datasets.

## Author Contributions

**Conceptualization:** Kurt G. Schilling, Justin Blaber, Adam W. Anderson, Bennett A. Landman.

**Data curation:** Kurt G. Schilling, Justin Blaber, Colin Hansen, Praitayini Kanakaraj, Tonia Rex, Susan M. Resnick, Andrea T. Shafer, Laurie E. Cutting, Neil Woodward, David Zald, Bennett A. Landman.

**Formal analysis:** Kurt G. Schilling, Colin Hansen, Bennett A. Landman.

**Funding acquisition:** Adam W. Anderson, Seth Smith, Tonia Rex, Andrea T. Shafer, Laurie E. Cutting, Neil Woodward, David Zald.

**Investigation:** Justin Blaber, Colin Hansen, Baxter Rogers, Seth Smith.

**Methodology:** Kurt G. Schilling, Justin Blaber, Colin Hansen.

**Resources:** Susan M. Resnick.

**Software:** Leon Cai.

**Supervision:** Bennett A. Landman.

**Validation:** Kurt G. Schilling, Leon Cai.

**Writing – original draft:** Kurt G. Schilling, Bennett A. Landman.

**Writing – review & editing:** Baxter Rogers, Adam W. Anderson, Seth Smith, Praitayini Kanakaraj.

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
