## [Decision Letter · Decision Letter 0]

19 Mar 2020

PONE-D-20-01149

Registration-free Distortion Correction of Diffusion Weighted MRI

PLOS ONE

Dear Mr. Schilling,

Thank you for submitting your manuscript to PLOS ONE. After careful consideration, we feel that it has merit but does not fully meet PLOS ONE’s publication criteria as it currently stands. Therefore, we invite you to submit a revised version of the manuscript that addresses the points raised during the review process.

We would appreciate receiving your revised manuscript by Apr 25 2020 11:59PM. To enhance the reproducibility of your results, we recommend that if applicable you deposit your laboratory protocols in protocols.io, where a protocol can be assigned its own identifier (DOI) such that it can be cited independently in the future. For instructions see: http://journals.plos.org/plosone/s/submission-guidelines#loc-laboratory-protocols

We look forward to receiving your revised manuscript.

Kind regards,

Pew-Thian Yap

Academic Editor

PLOS ONE

Journal Requirements:

2. Please update your Ethics statement to provide approval numbers for the studies under IRB approval. Please also include information on whether any data was collected specifically for this study, or whether all data used came from pre-existing datasets. Please ensure that this information is included both in the Ethics statement and in the manuscript itself.

3. Your ethics statement must appear in the Methods section of your manuscript. If your ethics statement is written in any section besides the Methods, please move it to the Methods section and delete it from any other section. Please also ensure that your ethics statement is included in your manuscript, as the ethics section of your online submission will not be published alongside your manuscript.

4. Please include your tables as part of your main manuscript and remove the individual files. Please note that supplementary tables (should remain/ be uploaded) as separate "supporting information" files

5. Thank you for stating the following in the Financial Disclosure section:

"This work was conducted in part using the resources of the Advanced Computing Center for Research and Education at Vanderbilt University, Nashville, TN. This work was supported by the National Institutes of Health under award numbers R01EB017230, and T32EB001628, and in part by ViSE/VICTR VR3029 and the National Center for Research Resources, Grant UL1 RR024975-01, and Department of Defense award number W81XWH-17-2-055. This research was conducted with the support from Intramural Research Program, National Institute on Aging, NIH. The content is solely the responsibility of the authors and does not necessarily represent the official views of the NIH. We gratefully acknowledge the support of NVIDIA Corporation with the donation of the Titan Xp GPU used for this research."

We note that you received funding from a commercial source: NVIDIA Corporation

6. We note that Figures 2, 3, 6, 8 in your submission contain copyrighted images. All PLOS content is published under the Creative Commons Attribution License (CC BY 4.0), which means that the manuscript, images, and Supporting Information files will be freely available online, and any third party is permitted to access, download, copy, distribute, and use these materials in any way, even commercially, with proper attribution. For more information, see our copyright guidelines: http://journals.plos.org/plosone/s/licenses-and-copyright.

1.         You may seek permission from the original copyright holder of Figure(s) [#] to publish the content specifically under the CC BY 4.0 license.

Reviewers' comments:

Reviewer's Responses to Questions

**Comments to the Author**

1. Is the manuscript technically sound, and do the data support the conclusions?

Reviewer #1: No

Reviewer #2: Partly

2. Has the statistical analysis been performed appropriately and rigorously? 

Reviewer #1: No

Reviewer #2: N/A

3. Have the authors made all data underlying the findings in their manuscript fully available?

Reviewer #1: Yes

Reviewer #2: Yes

4. Is the manuscript presented in an intelligible fashion and written in standard English?

Reviewer #1: No

Reviewer #2: Yes

5. Review Comments to the Author

Reviewer #1: The paper describes a learning-based distortion correction framework, where the structural T1 and distorted single-blip B0 images are used to synthesize undistorted B0 images. Later, the synthesized and distorted B0 images are fed into TOPUP for further correction. Lastly, FSL's eddy tool is used to do complete pre-processing of the DWIs.

Primary flaws of the paper are 1) presentation of the method, 2) novelty & technical soundness.

In Section 1, the authors highlighted the limitations of their previous work (Synb0-DisCo) but didn't show how these limitations are addressed by the proposed approach.

The title of the paper is misleading ("Registration-free"..), as the method relies on TOPUP and it is not truly registration-free. Also, I really don't get the idea of using TOPUP on top of B0 synthesis. If the synthetic undistorted B0 image is good enough then why run TOPUP?

It is difficult to understand the loss functions in textual form, so it would be better if the authors formulate all the loss functions mathematically.

All the figures are of poor quality. The text in the boxes of Fig. 4 is illegible. There is no network architecture shown. The surface overlays in Fig. 6 & 8 are also not clear.

Reviewer #2: The authors propose a novel method to predict an undistorted b0 from a distorted b0 and a structural image using deep learning (U-nets). This undistorted b0 is then used as a reference in FSL's topup to estimate the susceptibility field (i.e., fieldmap), which allows legacy data acquired without an alternative way to estimate the fieldmap to still be analysed. It's a well-presented and very promising approach to this problem.

My only issue is whether the algorithm will be useful in legacy datasets as this does require the U-net to generalise to other datasets. The authors claim in the abstract: "we show generalizability of the proposed approach to datasets that were not in the original training / validation / testing datasets". However, in the paper they only show 3 selected examples from 3 datasets, where the algorithm does indeed do a decent job. A more quantitative comparison of this claim could be easily produced from for example the distribution of the MI with the T1 (similar to what was used in the withheld dataset in the left panel in Figure 7). This would allow the authors to show whether the algorithm reliably generalizes to other dataset, which would be required before applying it to legacy datasets.

Minor issues:

1. Please give the units of the reconstructed voxel size in the BLSA dataset (page 3).

2. When mentioning the coregestration from b0 to T1 using epi_reg, it would be good to mention that this is a rigid-body transformation (6 degrees of freedom).

3. Was the T1 image registered to the standard after intensity normalization (implied by the text) or before intensity normalization (implied by Figure 3)?

4. There are many arrows missing and misplace in Figure 4. The ones I spotted are:

- There should be an arrow from b0_d_MNI_1 to the upper U-net (top part).

- There should be an arrow from b0_u_MNI to the top MSE (top part).

- There should be an arrow from b0_m_MNI_2 to middle MSE (top part).

- Why is there an arrow going upwards from T1_norm_MNI (top part)?

- In the lower part the arrow between b0_d_MNI_1 and the U-net is shifted upwards.

5. In the caption of Figure 6 there is no mention of which outline in the right panels is the distorted b0 and which is the undistored b0. This information is in the caption of Figure 8, but there it is the wrong way around (red is hopefully the outline of the undistorted b0). Also, in Figure 6, there are two typos, namely "With-held" should be "Withheld" and "let" should be "left".

6. What is the legacy synthetic distortion correction (S.D.) approach mentioned in the caption of Figure 7. It might be good to mention this in the results section when discussion Figure 7.

7. When discussing the timings in the discussion, it would be good to mention that the full pipeline of Synb0 still involves running topup, so the total runtime of Synb0 is much more than 2 minutes.

8. The mention of Figure 4 in the discussion should actually refer to Figure 6.

6. PLOS authors have the option to publish the peer review history of their article (what does this mean?). If published, this will include your full peer review and any attached files.

Reviewer #1: No

Reviewer #2: No

---

## [Author Response · Author response to Decision Letter 0]

21 May 2020

Please see Response to Reviewers document attached.

---

## [Decision Letter · Decision Letter 1]

2 Jul 2020

PONE-D-20-01149R1

Distortion correction of diffusion weighted MRI without reverse phase-encoding scans or field-maps

PLOS ONE

Dear Dr. Schilling,

Thank you for submitting your manuscript to PLOS ONE. After careful consideration, we feel that it has merit but does not fully meet PLOS ONE’s publication criteria as it currently stands. Therefore, we invite you to submit a revised version of the manuscript that addresses the points raised during the review process.

We look forward to receiving your revised manuscript.

Kind regards,

Pew-Thian Yap

Academic Editor

PLOS ONE

Reviewers' comments:

Reviewer's Responses to Questions

**Comments to the Author**

1. If the authors have adequately addressed your comments raised in a previous round of review and you feel that this manuscript is now acceptable for publication, you may indicate that here to bypass the “Comments to the Author” section, enter your conflict of interest statement in the “Confidential to Editor” section, and submit your "Accept" recommendation.

Reviewer #1: (No Response)

Reviewer #2: All comments have been addressed

2. Is the manuscript technically sound, and do the data support the conclusions?

Reviewer #1: Partly

Reviewer #2: Yes

3. Has the statistical analysis been performed appropriately and rigorously? 

Reviewer #1: Yes

Reviewer #2: N/A

4. Have the authors made all data underlying the findings in their manuscript fully available?

Reviewer #1: Yes

Reviewer #2: Yes

5. Is the manuscript presented in an intelligible fashion and written in standard English?

Reviewer #1: No

Reviewer #2: Yes

6. Review Comments to the Author

Reviewer #1: 1) Check if the statement is correct: "To account for this, the median value of the masked undistorted b0 was scaled such that it matched the masked median value of the undistorted b0". Distorted should be matched to undistorted B0 image?

2) In Fig. 7, what do different colored markers represent?

3) In Fig. 4, no input is mentioned at the beginning.

4) Language of the paper needs significant improvement. There are many grammatical errors, such as:

"Figure 8 shows that these can correct distortions on datasets that may differ from those the networks were trained on." 'These' should be changed to 'this'.

"The inverse transforms were used to convert the generated undistorted b0 back into subject space." Geometric transforms do not "convert" images; transforms deform/warp images.

Reviewer #2: The authors have addressed all my concerns.

I just have two minor comments:

1. In the intro of section 2 the authors state: "in order to provide topup the information necessary to perfectly correct the distorted diffusion data." The word "perfectly" is a bit of an overstatement (no tool is perfect), so I would suggest to remove it.

2. In the right panel in Figure 5 the dots are grey, not green as stated in the caption of Figure 5.

7. PLOS authors have the option to publish the peer review history of their article (what does this mean?). If published, this will include your full peer review and any attached files.

Reviewer #1: No

Reviewer #2: No

---

## [Author Response · Author response to Decision Letter 1]

3 Jul 2020

See Response to Reviewers document.

---

## [Editor Report · Decision Letter 2]

8 Jul 2020

Distortion correction of diffusion weighted MRI without reverse phase-encoding scans or field-maps

PONE-D-20-01149R2

Dear Dr. Schilling,

We’re pleased to inform you that your manuscript has been judged scientifically suitable for publication and will be formally accepted for publication once it meets all outstanding technical requirements.

Kind regards,

Pew-Thian Yap

Academic Editor

PLOS ONE
---

## [Editor Report · Acceptance letter]

10 Jul 2020

PONE-D-20-01149R2 

Distortion correction of diffusion weighted MRI without reverse phase-encoding scans or field-maps 

Dear Dr. Schilling:

I'm pleased to inform you that your manuscript has been deemed suitable for publication in PLOS ONE. Congratulations! Your manuscript is now with our production department. 

Kind regards, 

on behalf of

Dr. Pew-Thian Yap 

Academic Editor

PLOS ONE